# Low-Temperature Triple-Capillary Cryostat for Ice Crystal Growth Studies

Brian D. Swanson[1] and Jon Nelson[2]

[1]ESS Department, University of Washington, Seattle, WA USA , Laucks Foundation Research, Salt Spring Island, BC Canada
[2]Redmond Physical Sciences, Redmond, WA USA; E-mail: jontne@gmail.com

**Correspondence:** brian@laucksfoundation.org

**Abstract.** Ice crystals come in a remarkable variety of shapes and sizes that affect a cloud's radiative properties. To better understand the growth of these crystals, we built an improved capillary cryostat (CC2) designed to reduce potential instrumental artifacts that may have influenced earlier measurements. In CC2, a crystal forms at the end of one, two, or three well-separated, ultra-fine capillaries to minimize both potential crystal-crystal and crystal-substrate interaction effects. The crystals can be
initiated using several ice-nucleation modes. The cryostat has two vapor-source chambers on either side of the growth chamber, each allowing independent control of the growth chamber supersaturation. Crystals can be grown under a range of air pressures, and the supersaturation conditions in the growth chamber can be rapidly changed by switching between the two vapor-source chambers using a sliding valve. Crystals grow fixed to the capillary in a uniform, stagnant environment and their orientation can be manipulated to measure the growth rate of each face. The high thermal-mass of CC2 increases the stability and uniformity
of the thermodynamic conditions surrounding the crystals. Here we describe the new instrument and present several sample observations.

## 1  Introduction

Ice crystals are important in the radiation balance of the Earth's climate system (Liou and Yang, 2016; Heymsfield et al., 2017).
But we still lack knowledge of both the crystal-shape distribution in ice clouds and the processes responsible for the observed variation in crystal shapes. Previous studies have used a variety of techniques to grow ice crystals under simulated tropospheric conditions, but each experiment seems to give different normal growth rates (i.e., rate of face advancement normal to itself), even under similar conditions and using similar techniques. For example at -15°C see Beckmann and Lacmann (1982); Lamb and Scott (1972); Sei and Gonda (1989a, b); Gonda et al. (1994); Libbrecht (2003) and at -30°C see Kobayashi (1965); Sei
and Gonda (1989a); Gonda and Koike (1983); Gonda et al. (1994); Libbrecht (2003). Why is this?

At low supersaturations, some of the variability is likely due to crystal defects as the ice nucleation process is expected to leave each crystal facet with a different defect structure. However, as described in Nelson and Knight (1996), the variations may

also be caused by potential instrumental artifacts. For example, in growing crystals on a flat substrate (e.g., Shaw and Mason (1955); Hallett (1961); Lamb and Scott (1972); Beckmann and Lacmann (1982); Sei and Gonda (1989a)), the substrate-crystal edge could be a preferred site for new layer nucleation that does not exist without the substrate. Such substrates can also have epitaxial-induced strain effects (Cho and Hallett, 1984a, b), and the temperature gradients in the crystal can greatly reduce the

5 growth rates over those predicted assuming equal temperatures of crystal and substrate (Nelson, 1993). Growth on fibers can have smaller, yet still significant substrate effects. For example, images of small crystals grown on a thin fiber by Kobayashi (1958, 1961) show the fiber often exiting at a crystal corner or edge, which could be showing substrate-induced control over the crystal aspect ratio, but without the capability of rotating the fiber, one cannot rule out the possibility of fiber influence on the other cases as well. Similar questions regarding control of habit by the fiber can be seen in the small crystals in Bailey

and Hallett (2004). When the crystals grow away from the fiber, as in the larger crystals in Bailey and Hallett (2004), growth may occur on only one side of the fiber and the crystals may be close enough together to impede each other's growth rates through the vapor-density field (Westbrook et al., 2008). Neither effect typically occurs for cloud crystals as the number density, during the vapor growth phase, typically ranges from 0.1 to 10 crystals per $cm^3$ (Mace et al., 2001; Kärcher and Strom, 2003) so average crystal-crystal separation is typically mm (or more) in scale. In addition, many apparatuses have temperature and

supersaturation gradients within the chamber that make calculating the precise thermodynamic conditions difficult.

Several support-free methods were developed that reduce the potential for crystal-substrate interaction effects but they can have other issues. In vertical wind tunnels and cloud chambers (i.e., Yamashita (1973, 1974); Gonda (1980); Takahashi and Fukuta (1988); Takahashi et al. (1991)), it is difficult to control the growth conditions precisely. Also here the crystal seeding, which typically occurs near the top of the chamber, leads to crystal fall motions that makes it difficult to continuously monitor

the growth of individual crystal faces throughout the experiment. Electrodynamic levitation methods avoid potential crystal-crystal interaction effects but the rapid motion of the crystals makes high-clarity imaging from a variety of crystal orientations difficult (Swanson et al., 1999; Bacon et al., 2003; Magee et al., 2006; Harrison et al., 2016) . All support-free methods have the potential disadvantage of ventilation factors that can enhance the crystal growth rates of oscillating crystals. Finally, wall-ice formation is a general concern in most laboratory experiments because it can lower the supersaturation ($S$) in the chamber and

be unnoticed by an experimenter without an independent method of following $S$ throughout the experiment.

To observe crystal growth at low temperatures while minimizing such instrumental shortcomings, we built a new instrument called capillary cryostat 2 (CC2). The design is an improvement over the capillary device in Nelson and Knight (1996), hereafter CC1, in which the ice crystal grew at the tip of an ultra-fine glass capillary. Like the earlier device, CC2 practically eliminates temperature gradients, greatly reduces substrate effects, and allows all crystal faces to be monitored in a highly controlled,

uniform environment. But in addition, CC2 allows experimenters to follow the growth of, and possible interactions between, several crystals growing under identical conditions. It has two vapor-source chambers for making rapid supersaturation changes and for independent temperature and supersaturation control, and has an associated vacuum system and gauges for control of the growth chamber air pressure. To date, CC2 has proven useful for studying the formation and behavior of air pockets in ice (Nelson and Swanson, 2019).

## 2 New CC2 Instrument and Methods

The design is basically a 'box within a box within a box' (see Figs. 1 and 2). At the center is the 13 cm x 7.5 cm x 20 cm experimental chamber EC with its 3 chambers - the growth chamber is near the middle with vapor-source chambers both above and below. Figure 2 shows the growth chamber GC containing the growing or sublimating crystals of interest. The crystals sit on the ends or sides of three well-separated, pure-silica glass capillaries that extend down about 3-cm from the ceiling of the GC. Individual or multiple ice crystals can be suspended on each capillary. The upper and lower vapor-source chambers VSC control the humidity within the GC. A sliding valve blocks one or the other VSC from the GC. An actuator mechanism attached to the sliding valve allows the experimenter to select which VSC is actively setting the GC humidity. Inside each VSC is a vapor source VS mounted on top of a thermoelectric cooler TEC module. Each VS is typically filled with frozen high-purity liquid-chromatography HPLC water. The supersaturation or sub-saturation conditions inside the VSC are controlled by the temperature ($T_{VS}$) of the VS. Surrounding the EC is an optically clear liquid-cryogenic fluid (typically methanol or a silicone fluid) contained within the bath box. The bath box itself is surrounded by the vacuum-shroud box. A turbo-molecular pump typically evacuates the vacuum-shroud box to less than $10^{-5}$ torr to isolate and insulate the EC from the laboratory environment. The EC, bath box, and vacuum-shroud box have silica windows front and back for illumination and imaging of the ice crystals. The imaging is done (at a working distance of about 80 mm) with a choice of back or front illumination and Nikon SLR cameras attached to Leica tele-microscopic zoom lenses. The EC and VS temperatures and pressure are monitored using a LabView data acquisition program, HP switch/multiplexer, and a 5 1/2 digit digital multi-meter.

### 2.1 Instrumental issues addressed by CC2 design

We now describe how the CC2 design addresses several potential instrumental issues and how the capillary method can be used to obtain reliable data sets.

#### 2.1.1 Temperature stability and gradients in the instrumental chamber

Small changes in crystal temperature can have large effects on ice crystal shape. Near liquid-water saturation, a few °C change near $-8$°C changes long columns into thin tabular crystal forms (Takahashi et al., 1991). At low supersaturations, small temperature changes may significantly effect facet-normal growth rates since this rate can depend exponentially on the vapor source temperature $T_{VS}$ (via its control of supersaturation) when the face is free of new-layer-generating defects (i.e., "perfect" faces, which were commonly found in CC1).

The stability of two temperatures, that of the experimental chamber $T_{EC}$ and that of the ambient air surrounding the crystal $T_a$, are important. The temperature stability of the EC is set by 1) the cryogenic refrigerant temperature control from a Neslab ULT-80 bath-circulator unit (temperature stability exceeds $0.1$°C over a 3-hour period), 2) the room temperature stability, and 3) the large thermal mass of the EC (which smooths short-term temperature fluctuations via its roughly 15-min response time). Specifically, the EC was milled from a single block of tellurium-copper, then nickel plated on the outside and gold plated inside for surface uniformity and to reduce the potential for oxide formation and contamination. Temperature variations in $T_a$ will be

due to temperature fluctuations and gradients in the internal walls of the GC. A time-series measurement of the 12 thermistors buried in the walls of the EC show that the maximum fluctuation of the EC block is less than 50 mK over a 1-day period. We worried that the TECs might induce small gradients in the EC temperature but find no measurable additional thermal gradient in the GC when the TEC current $I_{TEC} < 0.5A$ - a value much larger than is needed for growth or sublimation conditions in a cold cloud. For a typical 11-hour period, the maximum gradient across the EC block was less than 10 mK. So we are comfortable assuming that $T_{EC} = T_a$ to within a few mK.

### 2.1.2   Precise control and stability of supersaturation around the crystal

While chamber temperature is relatively straightforward to measure and control, precise supersaturation measurement within a chamber along with measurement and control near the surface of a growing crystal is much more difficult. In any experiment we are concerned with both spatial and temporal gradients in the growth chamber supersaturation $S_{GC}$. The "gold standard" for crystal growth experiments involves two parts: a stable, controlled and gradient-free $S_{GC}$ within the experimental chamber, and the measurement of $S_a$ near the the surface of the growing (or sublimating) crystal at the same time the crystals are growing (or sublimating). No ice crystal growth rate experiment to date satisfies these 2 conditions. But the method used in CC2, although not yet at the "gold standard" level, has enhanced thermal stability and a relative gradient-free nature that is a large improvement over previous methods.

Within CC2 all crystals grow simultaneously within an approximately 1 cc volume near the center of the GC. Simultaneous growth in a chamber without gradients means all crystals experience the same thermodynamic conditions. Sequential growth experiments cannot claim all crystals experienced "the same" thermodynamic conditions without an actual measurement of $S_a$. Previous experiments have assumed gradient-free conditions and in this case the ambient supersaturation at the crystal surface given in % by

$$S_a = \frac{N_{eq}(T_{VS}) - N_{eq}(T_a)}{N_{eq}(T_a)} * 100, \tag{1}$$

where $N_{eq}$ is the equilibrium vapor density in #molecules/m$^3$ and $T_a$ is the temperature at the crystal surface. It is possible that other factors affect $S_a$ so in future experiments we will test the use of this equation. Other than the effect of thermal gradients within the EC (which are minimal as discussed above), the potential gradients in $S_{GC}$ within the GC can come from 3 potential sources: a) thermal instability of $T_{VS}$, b) gradients within the VS itself, and c) the presence of other ice crystals within the EC.

a) **Thermal stability of $T_{VS}$.** The numerator in eq. (1) is to first order proportional to the ice surface-temperature elevation $\Delta T = T_{VS} - T_a$. Thus, the relative uncertainty in supersaturation $\delta S_a / S_a = \delta \Delta T / \Delta T$. With feedback TEC temperature control the VS temperature standard deviation $\Delta T$ is typically about 3 mK (the variation observed over several hours). This gives an estimated uncertainty in $S_a$ of about $0.03\%$. To understand the meaning of $0.03\%$ supersaturation, consider that an ice crystal growing at the maximum possible rate at $-30°C$ at this supersaturation is about 60 $\mu$m per hour in a pure vapor (from the Hertz-Knudsen equation (e.g., eq. 1 of Holyst et al. (2015). This equation assumes a rough surface, or $\alpha = 1$. ), then we expect the uncertainty to add about 0.2 $\mu$m per hour additional growth to a 100-$\mu$m diameter spherical crystal in an atmosphere

of air (Maxwell's expression or Hertz-Knudsen divided by the vapor-diffusion impedance). This uncertainty is less than the measurement resolution over several hours growth.

b) **Thermal gradients within the VS.** Each VS consists of a gold-plated copper disc, machined such that the top portion forms a cup-shape that holds up to 2 g of water. Each VS has an exposed surface area of about 6 cm$^2$. As the typical grown crystal is less than 0.05-cm across, the VS surface area is usually more than 1000 times larger than the crystal being studied, and, in the absence of other crystals or wall ice, $S_a = S_{VS}$. This supersaturation is determined by the vapor-source temperature, which is controlled by setting $I_{TEC}$. The design improves upon that used in CC1 (Nelson and Knight, 1996) which used the solute method alone, although solutes can be used in the VSC as well.

Consider now the temperature difference between the thermistors imbedded in the VS cup $T_{VS}$ and the surface of the VS ice $T_{VSS}$. In general, we need this difference to be much less than the set temperature rise of the VS cup over that of the environment $\Delta T$; otherwise, our inferred supersaturation will be too high. To estimate $\frac{T_{VS}-T_{VSS}}{\Delta T}$, assume a steady state in which the rate of latent-heat loss at the source-ice surface (during a crystal-growth experiment) equals the sum of the i) rate of heat conduction through the VS ice plus ii) the heat loss from the surface to the surroundings. Consider just i) first. Assuming that the number of molecules of water leaving the VS ice per second equals the number depositing on the observed crystal on a capillary (i.e., steady-state), and using the Clausius-Clapeyron equation, this ratio can be shown to equal $\frac{R}{S_a}LA_r(\frac{k_B\beta^2}{\Omega\lambda})$, where $R$ is the surface-averaged normal growth rate of the crystal (i.e., normal to the surface), $L$ is the average thickness of the VS ice, $A_r$ is the ratio of areas between the observed crystal and the VS ice, $k_B$ is Boltzmann's constant, $\beta$ is the latent heat per molecule normalized by $k_BT$, $\Omega$ is the volume per molecule in solid ice, and $\lambda$ is the thermal conductivity of ice. This last factor in parenthesis involves only material properties and equals about $1x10^8$ s/m$^2$. Using $R = 5$ $\mu$m per hour, $S_a = 0.01$ (1%), $L = 1$ cm, and $A_r = 10^{-3}$, this ratio is $10^{-4}$. As these are roughly maximum values, we can generally assume the temperature offset to be negligible. (Also, as the factor depends on $L/\lambda$, the influence of temperature gradients in the vapor-source cup itself should be negligible due to the tellurium copper having a thermal conductivity nearly 200 times larger than that of ice (and $L$ being smaller).)

Concerning ii), the heat loss to the surroundings, there are four to consider: conductive loss from the ice to the air, conductive loss from the VS cup to the cup holder, convective loss to the air, and radiative loss to the walls. For the conductive loss to the air, we can estimate the effect by equating the heat flow through the ice to the heat flow through the air via conduction. The resulting temperature shift in the ice divided by ($T_{VSS}$ - $T_a$) (i.e., $\Delta T$ less the temperature shift in the ice) will equal the ratio of the conduction distances times the inverse ratio of the thermal conductivities. The first factor is about 1/3, and the second is about 0.015/2.1. Thus, this temperature shift is only about 1/300 of that of $\Delta T$ and can be ignored. The conductive loss from the VS cup to the cup holder would create gradients in the cup holder. However, the thermal conductivity of the Te-Cu cup is about 4000x that of the rubber o-ring holding it in place and nearly 40,000x that of the air gap. Thus, even though this gap is small, we can neglect the resulting thermal gradients in the cup.

The heat loss can also be convective if the vapor source is heated for growth experiments and requires a critical temperature difference between the ice surface and the top wall of the VSC. (If we instead use solute, as was done in CC1, then the issue cannot arise.) For growth experiments, the resulting convection may significantly cool the ice surface, so our aim is

to stay below the critical temperature. If we assume that the onset of convection occurs with a Rayleigh number of about 1500 (following Saxena et al. (2018), where this number is proportional to the cube of the chamber height, the temperature difference between wall and source, and properties of the air), then for our chamber and operating temperature, staying below this Rayleigh number requires the ice surface lie within about 0.5 K of the wall temperature. Finally, the influence of the radiative heat flux is considered in 2.1.4) below.

c) **The presence of other crystals.** We consider two cases separately: a few crystals very near the monitored crystal of interest and a large number of crystals on the wall as frost. The later case is discussed in 2.1.3 below; here we focus on the former. When other crystals are nearby a crystal of interest then $S_a$ can be less than $S_{VS}$ even when the area of the VS ice surface greatly exceeds that of any other ice in the system. In the case of the simultaneous growth of several observed crystals, the supersaturation near an observed crystal may be reduced due to the proximity to other crystals. Westbrook et al. (2008) estimate a 3-fold reduction in growth rate for close crystals along a fiber, a situation that simulated those in Bailey and Hallett (2004). At larger crystal separations, the effect has not been determined, but the $1/r$ dependence of the vapor-diffusion field away from a crystal suggests that to ensure a crystal-proximity effect of less than 10%, the crystals should be separated by nearly 10x their mean dimension. We find (result reported elsewhere (Swanson, 2019)) about a 30% reduction in facet-normal growth rate (caused by the vapor uptake by the neighboring polycrystal crystallites) for a prismatic crystal growing on top of a polycrystal as compared with similar prismatic crystals growing and separated by 100's of $\mu$m.

In CC2, the three capillaries are on non-parallel axes, and thus their separations are adjustable, allowing measurement of the proximity effect. They are easily set to be several centimeters apart, which is more than 100x their typical dimension of about 100 $\mu$m. For the growth of 3 ice crystals each less than 500 $\mu$m in size and separated by least 5 mm then we can assume $S_a = S_{VS}$ to within 10% of $S_{VS}$.

### 2.1.3 Frost formation on the experimental chamber walls

Frost can form on chamber walls and, if the area is large, can uptake a significant fraction of the source vapor. A highly controlled vapor-source supersaturation, $S_{VS}$, does not necessarily set the ambient supersaturation, $S_a$, at the center of the chamber where the crystals are growing if there is frost or condensate growing on the wall. Such frost is a particular problem when growing crystals sequentially because any measured difference in their rate or habit may not be inherent, but instead be due to their being affected more or less by frost. To reduce this issue, the CC2 windows allow observation of all surfaces inside the GC and VSC. But it is possible that this ice is so thin as to make it nearly invisible to the eye. An important factor is the relative surface area of the frost versus that of the VS ice. If their areas and thicknesses are the same, then the vapor-density in the EC would be mid-way between the equilibrium values for the VS ice and the chamber walls. However, the effect in practice would likely be worse because the frost layers would likely be much thinner than the VS ice, pushing the vapor-density closer to the equilibrium value for the walls due to the temperature-gradient effect in 3.1.2) above.

The windows on the sides of the VSC provide for easy detection of large frost crystals and, once noticed, that chamber can be immediately sealed off. In practice, we find that when frost crystals first appear, they are in the VSC, relatively close to the source ice. For the experiments described here, the VSC and GC internal walls were continuously monitored for frost. If frost

began to form in the attached VSC, then the sliding valve was changed to disconnect the VSC from the GC and the TEC in the other VSC was set to maintain the desired humidity in the GC. The ability to isolate one VSC from the GC when frost first occurs and to switch to a frost-free VSC allows us to continue to grow the crystals for long periods at near-constant $S_a$ conditions. To clear the frost off the walls of a VSC, we first evacuate the VSC and set its $T_{VS}$ to at least $10°C$ below $T_a$. The

5 frost typically left the walls within about 30 minutes under these conditions. For the conditions of the experiments described here, typically no frost was observed on VSC walls for at least 6 hours. The data set here was collected before frost started to form in the GC walls.

In a future paper, Swanson and Nelson (2019), we report results from droplet evaporation measurements done simultaneous with the crystal growth measurements. Measuring the evaporation rate of pure water droplets during crystal growth does give

a direct and independent measure of $S_a$ near the growing crystals surface. For these experiments one capillary is used to hold the evaporating droplet while the other two hold the growing or sublimating crystals. Results from these experiments demonstrate that accurately predicting $S_a$ at a chamber center requires careful calibration. For the results reported here we are concerned with facet-normal growth rate differences for crystals growing simultaneously under the same thermodynamic conditions. In these experiments the ice crystal surface area is small compared with that of the VS and no wall ice was present.

We continuously measure $T_{GC}$ and $T_{VS}$ and the enhanced thermal stability and control within CC2 gives us confidence that, within the variations caused by the measured temperature gradient across the EC, $S_a$ is to good approximation $S_{VS}$ within that 1 cc volume that contains the capillary tips. In future experiments, where a detailed comparison with crystal growth models is the goal, $S_a$ calibration measurements will be made along with the growth rate measurements.

### 2.1.4 Radiative heating effects

Radiative heating can occur in two places. First, consider thermal radiation between the VS surface and the VSC wall. The VS-GC temperature difference is typically less than $3.5°C$ - the value needed to achieve liquid-water saturation at $-40°C$. Due to the relatively small temperature differences involved, and also the very low emissivity of the gold plating of all interior walls, such a radiative heat transfer has a negligible influence on the VS surface temperature. Second, consider thermal radiation between the ice crystals and sources outside the windows. The ice crystals sit at $T_a$, but the thermal link is weak due to the

crystal being surrounded by air. Considering the different materials viewed by the crystal (windows, walls, and circulating fluid), determining the influence of radiative heating on ice-crystal temperature is best handled as an experimental issue. We examine this issue by monitoring the crystals in the growth chamber under controlled conditions in which the windows are alternately exposed or covered with low IR-emissive material. When we have tried this test, we observed no IR heating effects on the normal growth rates.

### 2.1.5 The effect of the capillary on crystal growth

The capillary holds the crystal steady, allowing clear imaging and viewing from several angles. Also, as the crystal starts at the capillary tip (typical case), one can usually measure the advance of all parts of the crystal with respect to the fixed capillary tip. Although these features are advantages of the method, the capillary can promote growth on one, two, or three faces.

Consider the examples in Fig. 3. The images show two crystals nucleated and grown at the same time, but on different capillaries, the left images from the front capillary Cf, the right images from the back capillary Cb. The crystal on the left (a, b) is nearly a hexagonal prism, but by measuring the distance from the capillary tip along the surface normal, one finds that the top right prism face has grown about 25% faster than the others. The capillary is seen exiting the crystal at the vertex
between this face and the top left prism. Subsequent images (not shown here) show the crystal growing larger, but the capillary remaining at this vertex. As the vertex stays at the capillary, these observations show that the capillary determines the relative normal growth rates of these two faces. The rotated view in (b) shows that the two basal faces have nearly equal normal growth rate, and neither is contacted by the capillary. Thus, the basal faces appear unaffected by the capillary. So, of the eight crystal faces, two are directly compromised by the capillary, but six are not directly influenced. In using data from this crystal, we
must consider the influence that the faster growth on the top two prism faces have on the vapor diffusion field near the other faces and the crystal temperature. In this way, the influence of the capillary may be overcome by crystal-growth modeling. The exact method will be described in a later publication.

The crystal on the back capillary, c-f in Fig. 3, shows further limitations and features. In this case, we cannot see the location of the capillary inside the crystal and must instead estimate its location by examining the growth sequence starting from
nucleation (not shown). Nevertheless, the basal-side views in (c) and (f) show that the capillary does not contact the basal faces (except possibly from the ice interior), yet one basal face grew faster than the other. Moreover, views (d) and (e), show that this crystal has two opposite prism faces that are much larger in area, and thus have much lower growth rates. The overall shape is similar to that proposed for crystals that generate the Parry arc (Westbrook, 2011).

Finally, consider the crystal in Fig. 4. In this case, the capillary exits at a corner, thus contacting one basal and two prisms.
As in other cases like this observed during both CC2 and CC1 experiments, the crystal does not start this way, but once the corner reaches the capillary, it always remains there (at least under constant conditions). Why does this occur? Clearly, the introduction of an interior glass-ice corner should promote new layer nucleation. If such a site is the most active on a given face, then that site will increase the normal growth rate of the face. Moreover, if this capillary is tilted towards a neighboring face, then the relative increase in growth rate over that neighboring face will bring the edge between the two faces closer to
the capillary. Once the edge reaches the capillary, it will stay there because the same promotion of layer nucleation will occur on the neighboring face. In three dimensions, if the capillary also tilts towards a third face, this process will bring that face to the capillary until the capillary exits at the common corner. However, if the layer-nucleation-promotion effect is relatively weak, then supersaturation gradients or a surface defect site may produce more rapid layer nucleation elsewhere, such at a nearby crystal corner. Thus, there are cases where, despite such promotion of layer nucleation, the face growth is controlled
by a more active site, making the capillary influence irrelevant. For example, if the capillary exits the crystal from near the face middle, it will likely lie at a lower-supersaturation region, with the more active step-generation site instead being at the corner. In such a case, the capillary may have a small local influence, but not influence the normal growth rates of the faces. These considerations also apply to crystals grown not at the tip, but midway along a capillary or any fiber. This explains why, for example, that in previous high-supersaturation experiments, the rapid-growing parts of the crystal are away from the fiber
(e.g., Nakaya (1954); Kobayashi (1958); Bailey and Hallett (2012)).

Occasionally, we observe indications of different influences from the capillary. For example, the crystal can appear to avoid contact with the capillary. This appears to be a vapor-shielding effect because it only occurs when the crystal size is within a few diameters of the capillary tip and only occurs where the crystal contacts the capillary. Other parts of the crystal are unaffected, and the effect vanishes when the crystal grows larger. Another effect that can occur is rapid growth up the capillary

in which the growth appears as smaller crystals of the same orientation. It is possible that this crystallization may be a result of thin-film crystallization on the glass capillary after a crystal nucleates at the tip. Both of these influences should be smaller with smaller-diameter capillaries, and may also be reduced with suitable coatings. Finally, within the ice just adjacent to the capillary (within a few capillary diameters), the interface may create strain effects in the ice. To date, we have not seen evidence of such effects, but they remain a possibility.

Thus, the capillary influence on new-layer production can be irrelevant in some cases and may be overcome using modeling in other cases, but should always be examined. Acknowledging this influence has two additional benefits. One, we may use it to study the nucleation process itself. Two, we can recognize the effect in other studies and realize that the resulting data may not be reliable. In future experiments, we plan to research these effects and develop strategies for quantifying their influence.

## 3 Growth of Snomax-Nucleated Crystals

Finally, we describe a case in which we nucleated crystals using a Snomax-water solution[1] and grew them for several days at low temperature and low supersaturation. Before insertion of the crystals the GC was prepared as follows: a) The internal GC walls were washed with acetone, ethanol, and finally with HPLC water. b) After window cleaning and re-assembly, the GC was flushed for over an hour with dry nitrogen and then the cooling began with the ULT-80 set to $0°$C. c) The VS cups were loaded with HPLC water and each TEC was set such that $T_{VS}$ remained about $-10°$C below $T_a$, d) A slow cool-down

was then initiated to the experimental temperature. Keeping the VS frozen with the above procedure avoided fogging of the GC windows and reduced the possibility of ice forming on the inside walls of the EC. Once the desired $T_a$ was established then $T_{VS}$ was set to produce the desired supersaturation $S_{VS}$ and the capillaries were inserted into CC2. Unlike the nucleation method used for the previous crystals, this one produced several crystals along each capillary. We report just on the ones at the ends of capillary Cf and Cr.

All crystals began the experiment as near-identical $\sim 20$ $\mu$m diameter liquid droplets of a Snomax-HPLC water solution. The 5 $\mu$m diameter capillaries were dipped into a Snomax solution made similar to Wood et al. (2002) and then inserted into CC2. The experiment was broken into two phases – Part A (which lasted 45 hours; $t = -45$ hrs to $t = 0$) directly followed by part B (which lasted 47 hours; $t = 0$ to $t = 47$ hrs). For the entire 92 hours the growth chamber temperature $T_a$ was held at $-29.8°$C. During part A, the crystals grew from the 20 $\mu$m frozen droplets into a variety of crystal shapes. During part A

the crystals grew for 28.5 hrs with $S_{VS} \sim 5\%$; followed by 9.5 hrs of no-growth with $-0.5 < S_{VS} < 0.5\%$ (after which some facet edge-rounding was observed); followed by an 7 hour growth period with $S_{VS} \sim 2\%$. During part B $S_{VS}$ was maintained such that $T_{VS} = -29.3 \pm 0.4°$C resulting in $S_{VS} = 5\%$.

---

[1]Snomax is a common ice nucleant used for making artificial snow at ski resorts, and is a product of York International, Victor, New York, 14564.

Figure 5 shows the crystals at both $t = 0$ and $t = 47$ hrs. The images have been magnified and crystal sizes are shown in Fig. 6. We define $a(t)$ as the growth normal to the prism face and $c(t)$ as the growth normal to the basal face. The value of a and c are diameters in these two normal directions. (Thus, they include measurements normal to faces potentially influenced by the capillary as discussed above.) The aspect ratio $AR = c/a$. In the lower image of crystal Cf, we see that the upper basal face, which is contacted by the capillary, grew faster than the lower facet. But for crystal Cr, the lower image shows the two basal faces to have nearly the same normal growth rate, and neither is contacted by the capillary. Thus, for the basal faces, Cf has one that is possibly affected by the capillary, but Cr appears unaffected by the capillary.

Since at least Gonda and Koike (1983), we have known that prismatic crystal aspect ratios can be different for similar growth conditions. The case here is consistent with this finding. In particular, we also find that the crystals in Fig. 5 responded to $T_a$ and $S_a$ in different ways. Comparing crystal shape at the beginning ($AR_0$) and end ($AR_{47}$) of part B, we find for Cf, $AR_{47}/AR_0 = 0.62$, while for Cr, $AR_{47}/AR_0 = 1.61$. This illustrates that, under the same conditions, a crystal can grow more plate-like at the same time another crystal is growing more column-like. (Concerning the capillary influence - the potential promotion of growth on one basal for Cf and one or two prisms for Cr oppose this trend. Thus, it is likely this finding is not due to a capillary influence.) The relative growth rates shown in Fig. 6 for the two crystals are also quite different. Both crystals ended part A of the experiment with tabular habits. But the thinner plate (Cr with $AR_0 = 0.46$) grew during part B to be more columnar, while the thicker plate (Cf with $AR_0 = 0.72$) grew to be more tabular. This behavior is also clear from the relative growth rates (indicated generally by the slope of the curves) in the $\hat{a}$ and $\hat{c}$ directions. The curves in Fig. 6 are from a simple two-parameter fit to the data set. We see here that a simple $t^{1/2}$ parameterization (where $t$ is the time) for both $c(t)$ and $a(t)$ fits well. With or without capillary influence the growth of both crystals is well described by a similar parabolic growth model as has been found for spherical droplets (Fukuta and Walter, 1970). A more detail discussion of these results is reported elsewhere (Swanson, 2019).

Crystals in previous experiments were often grown sequentially, making it difficult to ensure the exact same conditions were reproduced. Moreover, in many cases, experimenters were unable to follow the development of each crystal, and each crystal face, throughout the growth process. In our experiments, several well-separated crystals can be grown simultaneously and experience the same thermodynamic conditions. The crystals remain prismatic during the multi-day growth period, but large variations in $AR$ and growth rate in the $\hat{a}$ and $\hat{c}$ directions are observed. Such variations in $AR$ turn out to be typical for prismatic crystals (Bacon et al., 2003; Swanson, 2019) and show control of growth shape is likely via defect-driven surface processes. By using CC2 to measure the growth of individual crystal faces for a wide range of conditions we will be able to quantify the variability of facet normal growth rates.

## 4 Conclusions

We have built a new instrument to measure high-precision growth rates of ice crystals and droplets at temperatures down to -60°C. Preliminary observations have shown the advantage of following individual faces of multiple crystals in the CC2 apparatus. With CC2, thermodynamic control is much tighter than has been reported for previous instruments. The ability to

grow multiple crystals under identical thermodynamic conditions, starting from their nucleation and following each face over long time periods, as well as being able to track and remove frost during the experiment, gives us confidence that differences in observed behavior can be distinguished from instrumental effects. We expect the method will be complementary to our substrate-free electrodynamic balance methods (Bacon et al., 2003). To check these results, future experiments that combine

5 both techniques are planned.

*Author contributions.* BS and JN designed and assembled the new instrument. BS and JN designed the experiments and carried them out. BS developed the Labview data acquisition code, made the figures and prepared the manuscript with contributions from JN.

*Competing interests.* The authors declare that they have no conflict of interest.

*Acknowledgements.* We thank Chris Foreman at Foreman CNC Machining LTD, Dave Shatford at Met-All-Fab in Sidney BC and Glenn

10 Ryan at Limited Productions Inc. Bellevue WA for assistance with the fabrication of the instrument. We thank Hawk Ridge Systems for providing a copy of SolidWorks CAD software. We thank Mary Laucks for helpful comments on this manuscript. This work is supported by the National Science Foundation grant AGS-1348238 and by the Laucks Foundation.

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

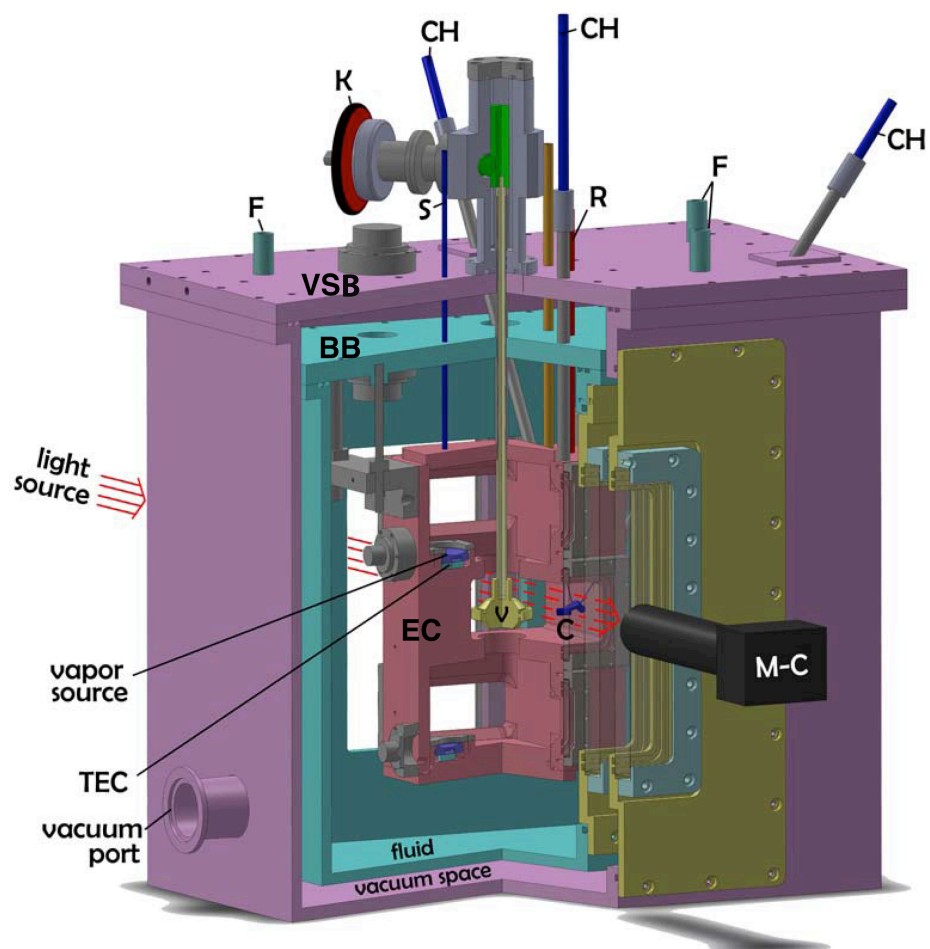

**Figure 1.** Cut-away of the triple-capillary cryostat (CC2). Both the (turquoise) bath box BB and (purple) vacuum-shroud box VSB surround the (red) experimental chamber EC. The growth chamber GC is the middle chamber in the EC where the crystals C are located. The top and bottom chambers are the two vapor-source chambers VSC each containing a vapor source VS situated on a thermoelectric cooler TEC. The setting of sliding valve V determines which VS sets the humidity in the GC. Other features are CH = capillary holders, F = cryogenic-fluid tubing, K = knob for sliding valve, S and R = tubing for filling vapor-source holders and for monitoring VSC pressure, and MC = SLR camera with telemicroscopic lens. Three sets of silica windows separate the laboratory air and the inside of the GC. For dimensions, the EC is 7" high and the CH tubes are 0.25" diameter.

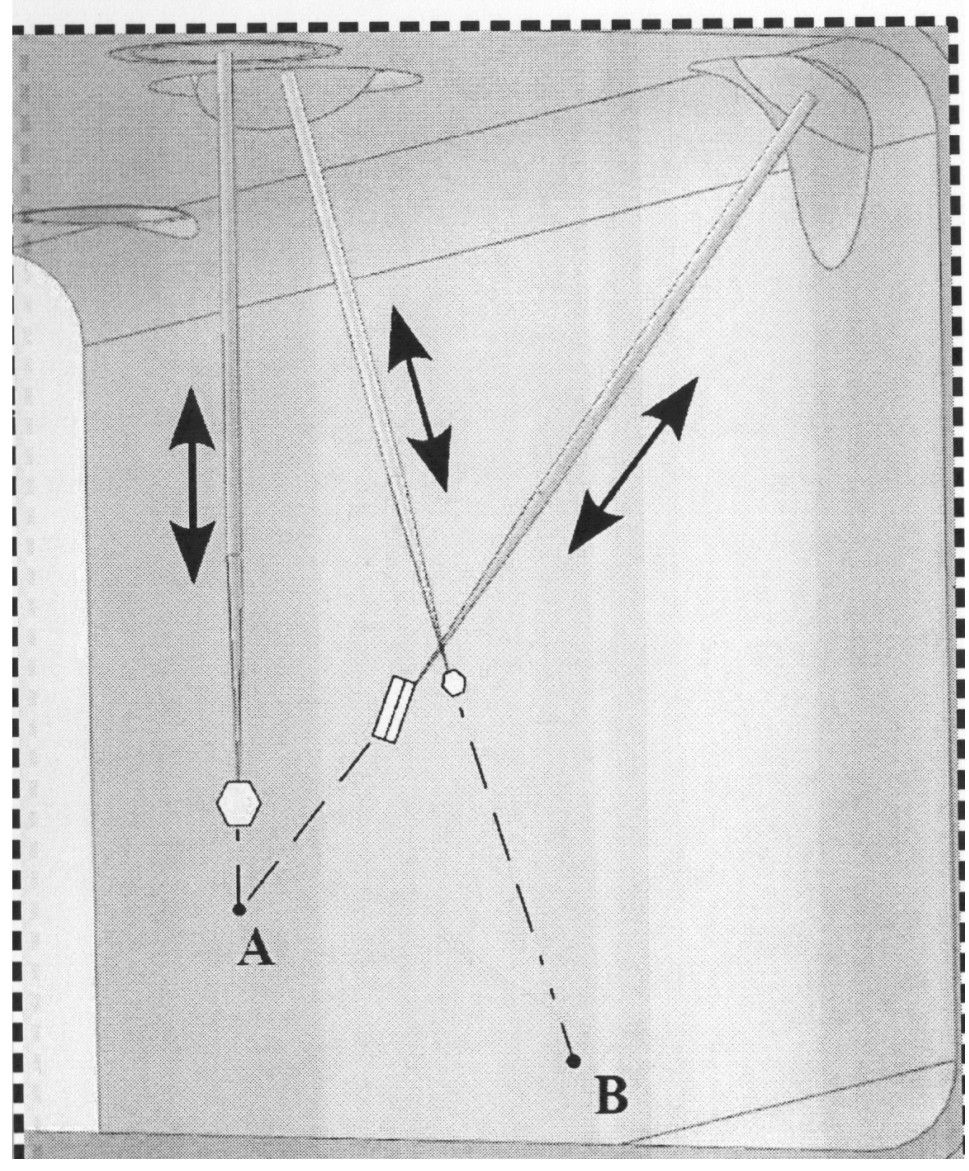

**Figure 2.** The three capillaries inside the growth chamber (GC) with ice crystals growing at their tips. From left, they are the front capillary Cf, the back capillary Cb (extends to point B on the front window), and the right capillary Cr (intersects with Cf at point A). Capillaries are positioned at center of the GC, typically with their ends within a 1 cc volume, and each capillary can be translated in and out or rotated $360°$.

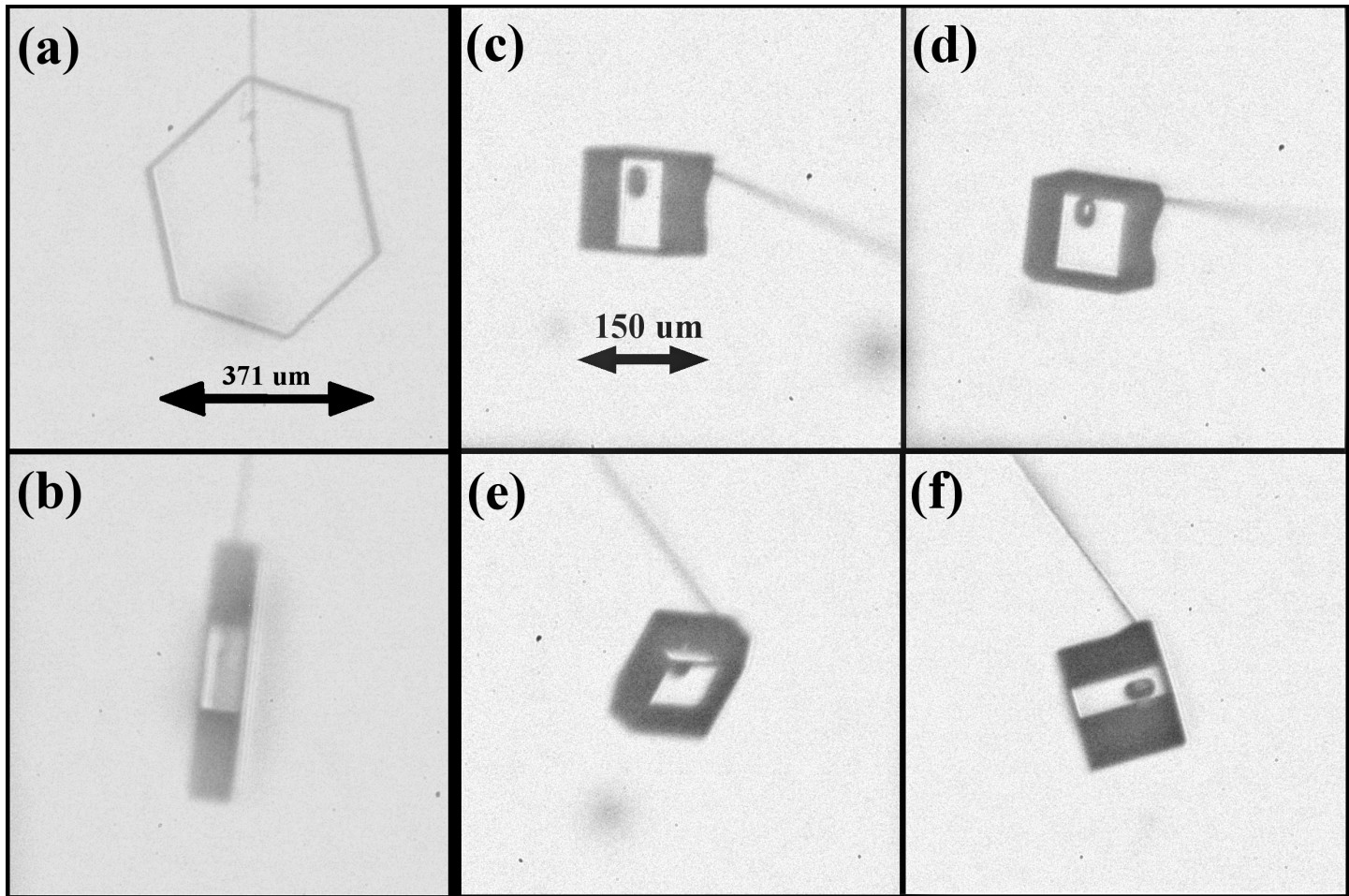

**Figure 3.** Crystals grown on capillary Cf and Cb. Both crystals nucleated and grew at the same time at -17 °C and about 1% supersaturation. a) Cf front view. b) Cf side view. c)-f) are four views of Cb where the difference in capillary direction is due to capillary rotation and the curvature of the capillary.

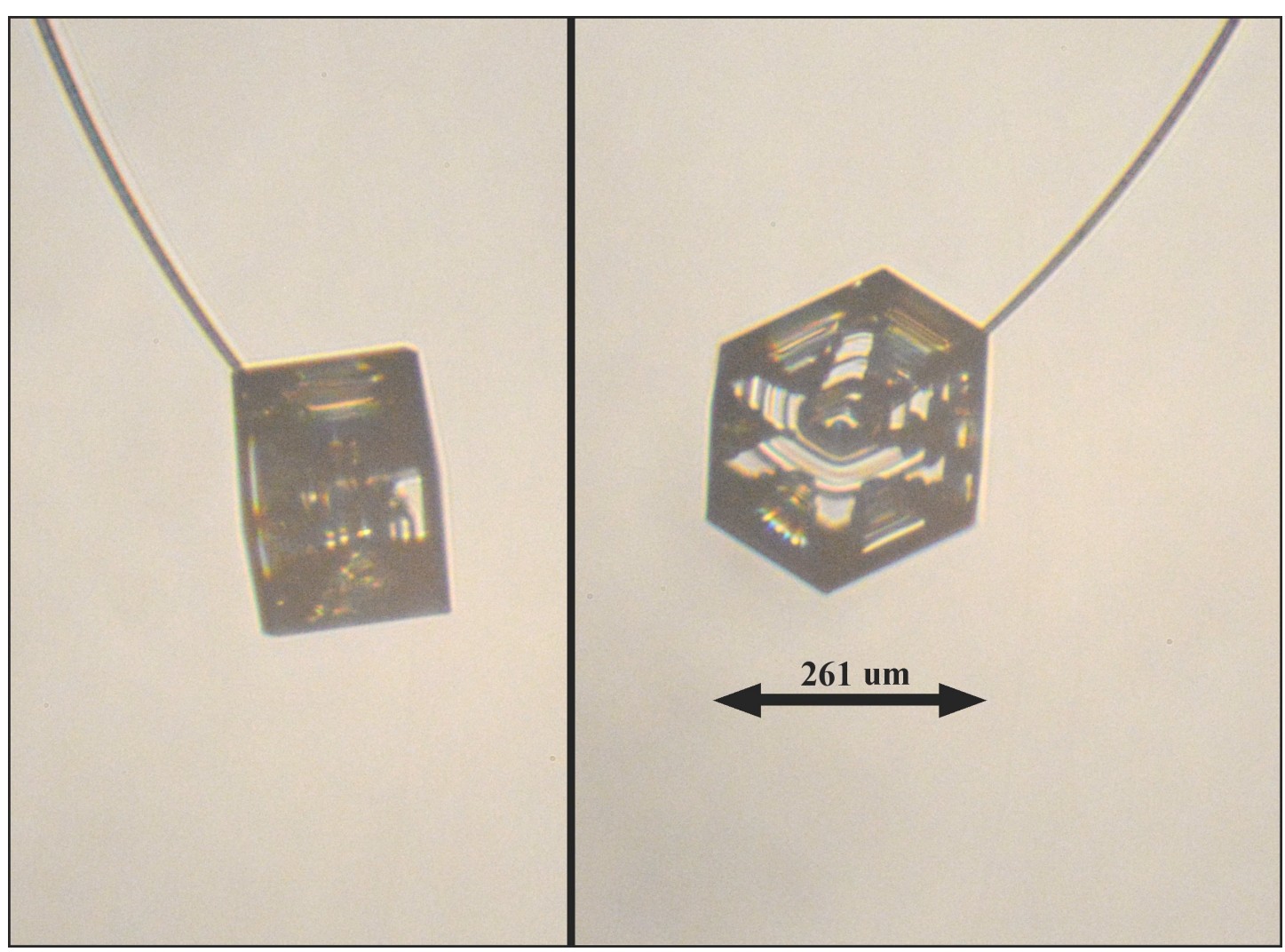

**Figure 4.** Side and front view of skeletal crystal grown on Cf.

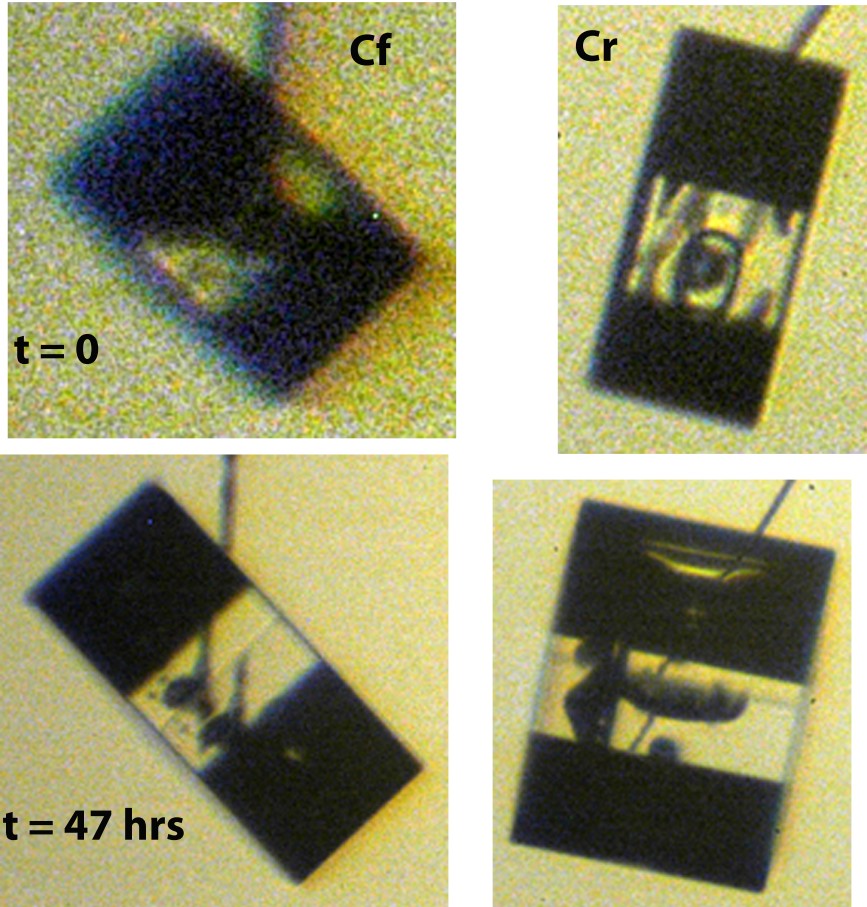

**Figure 5.** Crystals nucleated from Snomax particles on Cf (left) and Cr (right), grown simultaneously under the same conditions. Top row of images were taken at t = 0 and bottom row of images were taken at t = 47 hrs. Both crystals have a symmetric prismatic hexagonal shape but developed remarkably different aspect ratios. During part B of the experiment the Cf crystal decreased in aspect ratio while the Cr crystal increased in aspect ratio.

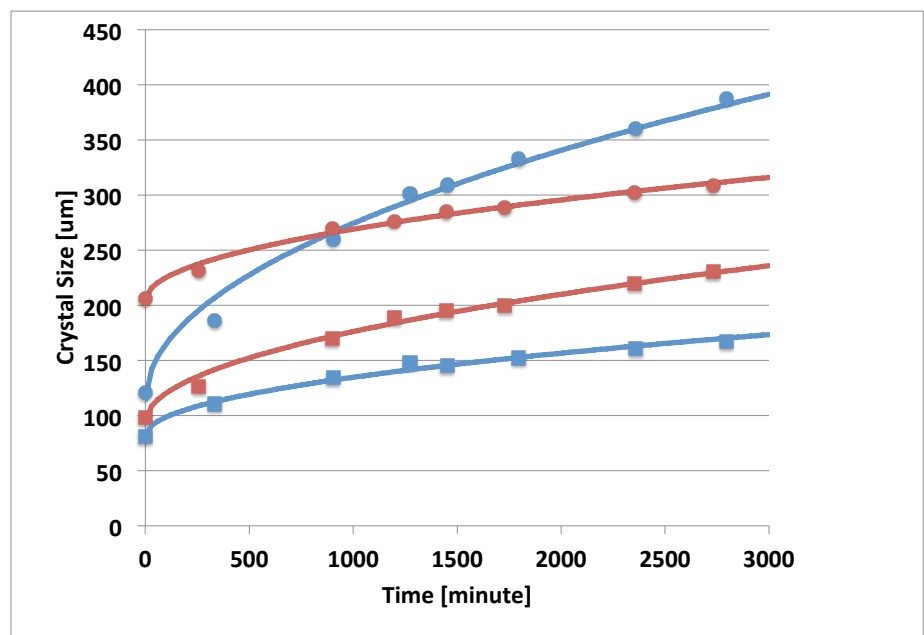

**Figure 6.** Crystal dimensions $a(t)$ (circles) and $c(t)$ (squares) measured during part B of the experiment for the crystals shown in Fig. 5. Blue points are for the crystal on Cf and the red points are for the crystal on Cr. The lines are the best fit for each crystal to a two-parameter power-law parameterization $a(t) = a_0 + g_a * t^{1/2}$ and $c(t) = c_0 + g_c * t^{1/2}$.