# Peer review of "Low-Temperature Triple-Capillary Cryostat for Ice Crystal Growth Studies"

_Atmospheric Measurement Techniques, 2019_

## Referee Comment (RC1) · Anonymous Referee #1 · 14 Aug 2019

This is a useful technical paper describing an advancement in the way that simulations of ice crystal growth in the laboratory are performed. A lot of work has gone into this chamber, and I'm happy to see all the considerations & analysis published so others can use it and understand the strengths and weaknesses of the technique.

I recommend publication, following some minor corrections.

Introduction "but each experiment seems to give different normal growth rates (i.e., rate of face advancement normal to itself), even under similar conditions and using similar techniques" - can you provide examples of this, and relevant citations here?

I felt section 2 was a very long unbroken section. It would benefit from being broken up a bit - for example splitting into subsections and including more of a "road map" at the

start of the section outlining the issues to be addressed

Equation 1 - I'd say \rho is more conventional notation for density... The analysis that follows could be spelled out more clearly. Why is the numerator proportional to dT?

You do a "back of the envelope" calculation here, with the Hertz-Knudsen equation - what assumptions does this calc make? e.g. regarding crystal + growth kinetics.

"If we assume that the onset of convection occurs with a Rayleigh number of about 1500,..." more background needed. can you justify this threshold, and define Ra physically

Page 5, last paragraph. Up to this point the anslysis seems to suggest that Sa can be estimated very precisely. But reading this last paragraph, I wasn't sure what to think. The authors conclusion needs to be more explicit here. You say the computed Sa in your "other experiments" was different to the real value (using droplet as a reference for the environmental saturation ratio). Can you be quantitative? How different? More than you would expect from the preceding estimates? If so, why might this be? And what is the implication for analysis of results from the chamber generally?

---

## Referee Comment (RC2) · Anonymous Referee #3 · 16 Aug 2019

review of "Low-Temperature Triple-Capillary Cryostat for Ice Crystal Growth" AMT manuscript 2019-137

This is a fairly well written description of a system for studying the growth of ice crystals in the atmosphere. How crystals grow and what determines their distribution of habit and size is a very important question for meteorology, and this paper represents significant progress in answering that question. I do have some comments on the paper however. If these are adequately dealt with, this paper definitely should proceed to publication in the journal.

Page 2 line 10; there is the statement "Neither effect typically occurs for cloud crystals". This needs some substantiation, at least in regard to the proximity of other growing crystals. Could the authors provide an estimate of the concentration of ice nuclei in a

typical cloud?

Section 2. This section purports to list several issues, and how they are solved in the CC2 design. The latter part of this aim seems to have been forgotten by the time point 5 is reached - there is plenty of discussion of the issues associated with capillaries interacting with crystal faces or vertices, but this is not tied to the CC2 design. This section would also be easier to follow if it were organised with subsections, rather than a list.

Section 3. Snowmax is apparently a trademark? A reference to a supplier (or a recipe when the name is first used) should be provided.

Reference list; the two references to Swanson and Nelson (2019 a,b) are quite inadequate!

Another very minor point is in the opening sentence of the second paragraph (of section 1) the authors do seem to like the work "likely" overmuch.

---

## Referee Comment (RC3) · Anonymous Referee #2 · 22 Aug 2019

Referee report on the manuscript "Low-Temperature Triple-Capillary Cryostat for Ice Crystal Growth Studies" by Swanson and Nelson.

The manuscript by Swanson and Nelson describes a steady state diffusion chamber designed for high-precision studies of ice crystal growth and sublimation. It appears to be a companion paper to the study of the formation of air pockets in growing ice crystals (Nelson and Swanson 2019) which has been conducted with the apparatus described here. The manuscript presents a very thorough description of the apparatus and discuss deeply the principles of operation and potential error sources. The images of the ice crystals grown with the help of the apparatus are amazing and obviously demonstrate the ability of the system to maintain stable temperature conditions for a very long time.

The manuscript, however, provides no convincing evidence that the apparatus can be used for studying diffusion growth of ice crystals under *predictably controlled* supersaturation conditions. By that I mean that in order to understand and to describe the crystal growth, the water vapor pressure in the vicinity of the crystal has to be set to and precisely maintained at the *predefined* value, which can be either derived from the instrumental parameters or obtained via calibration. This ability has not been demonstrated in the manuscript. Instead, there is a lot of discussion of the potential errors and why they should have negligible effect on the growth rate of the crystal. What I am desperately missing is the characterization of the instrument in terms of supersaturation as a function of a) temperature of the growth chamber,  b) temperature of the both vapor sources, c) spatial coordinate in the growth chamber, d) time. As authors themselves put it: "*We conclude that accurately predicting and maintaining a constant $S_a$ at a chamber center without a direct measurement of the supersaturation requires careful calibration*" (page 5 line 34-35), but the calibration is missing. In fact, in the whole manuscript, not a single value of the supersaturation (or vapor pressure) in the growth chamber is given. The closest occasion where the word "supersaturation" is used in conjunction with any numerical values is "*During part A $S_a$ was not highly controlled but conditions were maintained such that −0.5% < $S_a$ < 0.5%. During part B $S_a$ was controlled such that $T_{VS}$ = −29.3±0.4 °C.*" (page 8 lines 30-31).  How this value of $S_a$ has been deduced? Why were the crystals growing if the supersaturation was zero on average?

The explanation why the actual supersaturation cannot be derived from the temperature of VS is offered on page 6, starting from line 1: "*In a highly controlled vapor-source supersaturation, $S_{VS}$, does not necessarily set the ambient supersaturation, $S_a$, at the center of the chamber where the crystals are growing if there is unobserved condensate growing on the wall.*" The issue is being addressed by observing the chamber surfaces visually, with the remark "*But it is possible that this ice is so thin as to make it nearly invisible to the eye*" (page 6 lines 6-7) . I don't see how one can control AND measure the actual supersaturation under these conditions.

Now, the authors claim that "…, *typically no frost was observed on VSC walls for at least 6 hours*" (page 6 line 20). In this case the question arises, how it was possible to conduct an experiment for 92 hours as described in the section 3? Obviously, this would require multiple de-icing steps as described on page 6, lines 18-19, during which the GC has to be disconnected from the VSC and reconnected to the second VSC with the VS set exactly to the same temperature. Or was the GC just left connected to the VSCs resulting in no supersaturation, as implied by the sentence "*During part A $S_a$ was not highly controlled but conditions were maintained such that −0.5% < $S_a$ < 0.5%.*"? If the setup was build to study the ice crystal growth at atmospherically relevant conditions, it should be possible to set and maintain supersaturations up to 30%. Nothing in this manuscript tells me that this is feasible.

If calculation of the supersaturation based on the instrumental parameters is not possible, a calibration can be achieved by measuring diffusion growth or evaporation of a reference particle – droplet of a

known solution. Apparently, the authors have done that: "…*we have in addition used droplet evaporation measurements done simultaneously with crystal growth measurements to give a direct, and independent, estimate of $S_a$*" (page 5 line 30). It's disappointing that this direct and independent estimate of the supersaturation is neither given nor discussed further. The only measurements of the crystal growth rate presented in the manuscript (Figure 6) are not compared with any other measurements or with theory; we learn that the growth rate can be fitted by a two-parameter power-law parameterization, but no further attempt of interpretation is given. Actually, even the fit parameters are not given or discussed, and the reader is informed "*A more detail discussion of these results is reported elsewhere*" leading to a reference (Swanson 2019a) that has a different title and dedicated to a different topic (I assume this is the reference to the paper in ACPD by the same authors. Actually, the reference to Swanson 2019b leads to nowhere). I have not been able to find any discussion of these results in the companion paper.

I am afraid I cannot recommend publishing the paper in its present form. It should be thoroughly revised aiming at providing a verifiable characterization of the apparatus under wide range of experimental conditions. If the supersaturation cannot be calculated from the instrumental parameters, it should be calibrated in a dedicated experiment with evaporation or condensation of inorganic solution droplet, and the results reported together with the theoretical model used for the simulations. I am not in a position to give advice on the chamber design, but perhaps it would be better to create well-defined wall boundary conditions in the VSC and CG chambers by covering walls with ice than relying on the absence of water adsorbed on the bare metal walls of the chamber.

1.      Nelson, J. and B. Swanson (2019). "Air pockets and secondary habits in ice from lateral-type growth." Atmos. Chem. Phys. Discuss. **2019**: 1-51 doi: 10.5194/acp-2019-280.

---

## Author Comment (AC1) · 26 Sep 2019

**Reply to the reviewers**

**Anonymous Referee #1**

This is a useful technical paper describing an advancement in the way that simulations of ice crystal growth in the laboratory are performed. A lot of work has gone into this chamber, and I'm happy to see all the considerations & analysis published so others can use it and understand the strengths and weaknesses of the technique.

I recommend publication, following some minor corrections.

Our Reply: We thank the reviewer for their helpful suggestions. We point out here the changes made to the revised manuscript:
* * *
Introduction "but each experiment seems to give different normal growth rates (i.e., rate of face advancement normal to itself), even under similar conditions and using similar techniques" - can you provide examples of this, and relevant citations here?

Our reply: We followed the reviewer's suggestion and have added citations to the data sets showing the large variations at -15C and -30C.
* * *
I felt section 2 was a very long unbroken section. It would benefit from being broken up a bit - for example splitting into subsections and including more of a "road map" at the start of the section outlining the issues to be addressed

Our reply. We have added new subsection headings in bold to increase the readability of this section.
* * *
Equation 1 - I'd say \rho is more conventional notation for density. . . The analysis that follows could be spelled out more clearly. Why is the numerator proportional to dT?

Our Reply: Equation 1 is a standard definition of supersaturation. N is the common symbol for number density in the crystal-growth literature. We prefer the symbol N to avoid confusion as the symbol \rho is used for the mass density of the air and vapor. Also \rho is the usual symbol for the mass density of water and ice, \rho_w and \rho_i, respectively. The numerator is

equilibrium number density difference between the vapor source temperature and the surface temperature. We have rewritten this section of the paper to add clarification to the text.
* * *
You do a "back of the envelope" calculation here, with the Hertz-Knudsen equation - what assumptions does this calc make? e.g. regarding crystal + growth kinetics.

Our Reply: The estimate was made with the assumption of $\alpha = 1$. We have added a reference to the Hertz-Knudsen equation to the text.
* * *
"If we assume that the onset of convection occurs with a Rayleigh number of about 1500,. . ." more background needed. can you justify this threshold, and define Ra physically

Our Reply: We have estimated the Rayleigh number for the onset of natural convection. We have referenced an experimental and numerical study showing approximately this value for convective onset and have added a little more explanation to the text.
* * *
Page 5, last paragraph. Up to this point the analysis seems to suggest that Sa can be estimated very precisely. But reading this last paragraph, I wasn't sure what to think. The author's conclusion needs to be more explicit here. You say the computed Sa in your "other experiments" was different to the real value (using droplet as a reference for the environmental saturation ratio). Can you be quantitative? How different? More than you would expect from the preceding estimates? If so, why might this be? And what is the implication for analysis of results from the chamber generally?

Our Reply: We have rewritten this section to explain more clearly the limits to $S_a$ determination. The "gold standard" for crystal growth experiments is stable T and S conditions in a chamber along with direct measurements of S and T near the crystal surface simultaneous with the growth rate measurements. To date there have been no experiments with a direct measurement of S near the crystal surface simultaneous with the growth rate measurements. We have made droplet evaporation measurements simultaneous with crystal growth measurements to obtain a direct measurement of $S_a$. But precise determination of $S_a$ is not required here as we are focusing on the differences in facet-normal growth rates for crystals growing simultaneously in the chamber under the same growth conditions. We have added the estimated value of $S_a$ using Eq. 1 to the text. Unfortunately there isn't space here for describing in detail our new method for more precise $S_a$ determination and this will be reported elsewhere.

---

## Author Comment (AC2) · 26 Sep 2019

**Response to Referee #3**

This is a fairly well written description of a system for studying the growth of ice crystals in the atmosphere. How crystals grow and what determines their distribution of habit and size is a very important question for meteorology, and this paper represents significant progress in answering that question. I do have some comments on the paper however. If these are adequately dealt with, this paper definitely should proceed to publication in the journal.

Our Reply: We thank the reviewer for their helpful suggestions.  Here are the changes made to the revised manuscript:
* * *
Page 2 line 10; there is the statement "Neither effect typically occurs for cloud crystals". This needs some substantiation, at least in regard to the proximity of other growing crystals. Could the authors provide an estimate of the concentration of ice nuclei in a typical cloud?

Our Reply: We have clarified the text with references to typical number concentrations in ice clouds.
* * *
Section 2. This section purports to list several issues, and how they are solved in the CC2 design. The latter part of this aim seems to have been forgotten by the time point 5 is reached - there is plenty of discussion of the issues associated with capillaries interacting with crystal faces or vertices, but this is not tied to the CC2 design. This section would also be easier to follow if it were organized with subsections, rather than a list.

Our Reply: We have clarified the purpose of this section and added subsection headings in bold to increase the readability of this section.
* * *
Section 3. Snowmax is apparently a trademark? A reference to a supplier (or a recipe when the name is first used) should be provided.

Our Reply: We have added a footnote to the Snomax supplier.  The recipe for the Snomax solution used is discussed in Wood et al. 2002 and we have included this reference.
* * *
Reference list; the two references to Swanson and Nelson (2019 a,b) are quite inadequate!

Our Reply: This manuscript is one of the first describing experiments done in the new CC2 instrument.  Unfortunately all manuscripts describing the results from this work are not yet submitted for publication. In keeping with convention we have added "Unpublished Manuscript" to these references.
* * *
Another very minor point is in the opening sentence of the second paragraph (of section 1) the authors do seem to like the work "likely" overmuch.

Our Reply:  Indeed annoying….   We have rewritten the sentence.

---

## Author Comment (AC3) · 26 Sep 2019

**Response to Referee #2**

The manuscript by Swanson and Nelson describes a steady state diffusion chamber designed for high- precision studies of ice crystal growth and sublimation. It appears to be a companion paper to the study of the formation of air pockets in growing ice crystals (Nelson and Swanson 2019) which has been conducted with the apparatus described here. The manuscript presents a very thorough description of the apparatus and discuss deeply the principles of operation and potential error sources. The images of the ice crystals grown with the help of the apparatus are amazing and obviously demonstrate the ability of the system to maintain stable temperature conditions for a very long time.

Our reply.  We thank the reviewer for their careful review.  The reviewer has several concerns and we have rewritten the sublimation section of the paper to clarify how we determine S_a and we have included the S_a estimate in the text for the Section 3 result.
* * *
The manuscript, however, provides no convincing evidence that the apparatus can be used for studying diffusion growth of ice crystals under *predictably controlled* supersaturation conditions. By that I mean that in order to understand and to describe the crystal growth, the water vapor pressure in the vicinity of the crystal has to be set to and precisely maintained at the *predefined* value, which can be either derived from the instrumental parameters or obtained via calibration. This ability has not been demonstrated in the manuscript. Instead, there is a lot of discussion of the potential errors and why they should have negligible effect on the growth rate of the crystal.

Our reply.  We have rewritten this section of the manuscript to clarify how S_VS is determined. We hadn't realized the interest in the estimated S_a for simultaneous growth experiment but we now have provided the estimated S_a for the results presented here.

It seems that the previous version led to some confusion.  Our chamber is not a diffusion chamber.  Hallett and Mason used a diffusion chamber, as did Bailey and Hallett.   Ours is a different design.  The sliding valve seals one VS or the other from the GC during operation. So only one of the two VS is setting S_gc.  In our chamber the supersaturation is set by T_vs and T_gc and is given by Eq. 1.   This is the same method of setting S_gc used in numerous other studies such as the Gonda 1983 and 1994 studies.  But the thermal control in CC2 is much improved compared with previous methods.  Gonda 1982 and most other previous experiments do not report the size of their spatial and temporal temperature fluctuations whereas the CC2 design specifically minimizes these gradients.
* * *
What I am desperately missing is the characterization of the instrument in terms of supersaturation as a function of a) temperature of the growth chamber, b) temperature of the both vapor sources, c) spatial coordinate in the growth chamber, d) time. As authors themselves put it: "*We conclude that accurately predicting and maintaining a constant $S_a$ at a chamber center without a direct measurement of the supersaturation requires careful calibration*" (page 5 line 34-35), but the calibration is missing. In fact, in the whole manuscript, not a single value of the supersaturation (or vapor pressure) in the growth chamber is given. The closest occasion where the word "supersaturation" is used in conjunction with any numerical values is "*During part A $S_a$ was not highly controlled but conditions were maintained such that $-0.5\% < S_a < 0.5\%$. During part B $S_a$ was controlled such that $T_{VS} = -29.3 \pm 0.4\ °C.$*" (page 8 lines 30-31). How this value of $S_a$ has been deduced? Why were the crystals growing if the supersaturation was zero on average?

Our reply. CC2 is designed for experiments observing multiple crystals growing simultaneously on tiny capillaries all within about a 1 cc volume at the center of the EC. The temperature stability of the EC has been measured to be less than 50 mK over a 1 day period. The temperature gradient across the 13 x 7.5 x 20 cm EC is on order of 10 mK over a 11 hour period. So to first-order the supersaturation gradient across this volume is given approximately by $\delta S \approx (\delta T/T) * S$ which is negligibly small. Therefore S_vs can be estimated using Eq. 1 as stated in the manuscript on P. 4-7.

The paper shows several examples of the high-resolution imaging possible in the CC2 and detailed measurements of crystal size can be made using these images. In the Snomax nucleated crystal experiment (results shown in Figs. 5 and 6) multiple crystals were growing *simultaneously* on the capillaries all within about a 1 cc volume at the center of the EC. These results are showing the crystal shape differences as a result of simultaneous growth with all crystals experiencing the same supersaturation. We conclude supersaturation differences are not responsible for the differences observed here. The facet normal growth rate and shape differences are instead likely due to different growth processes occurring at the different facet surfaces. This is a small result but seemingly not well appreciated in previous experiments due to possibility of changing growth conditions surrounding the crystals of interest. S_vs for the experiment is reported near the bottom of P. 9.
* * *
The explanation why the actual supersaturation cannot be derived from the temperature of VS is offered on page 6, starting from line 1: "*In a highly controlled vapor-source supersaturation, $S_{VS}$, does not necessarily set the*

*ambient supersaturation, $S_a$, at the center of the chamber where the crystals are growing if there is unobserved*

*condensate growing on the wall.*" The issue is being addressed by observing the chamber surfaces visually, with

the remark "*But it is possible that this ice is so thin as to make it nearly invisible to the eye*" (page 6 lines 6-7) . I

don't see how one can control AND measure the actual supersaturation under these conditions.

Our Reply:  Previous experiments have simply estimated the growth temperature and supersaturation conditions in various chambers without detailed calibrations or actual measurements.  No previous ice crystal growth experiment has actually measured the temperature and supersaturation near the surface of the growing or sublimating crystal.  Such a result in itself would be a major breakthrough for crystal growth technology and the report of such an accomplishment is not our claim here.  Instead we expect S_vs is well approximated by Eq. 1 and deviations are due primarily to the temporal and spatial gradients in the growth chamber and vapor source chamber temperatures.  These gradients are described in detail in section 2.

For this report higher-precision determination of S_gc is not required as this is a simultaneous growth experiment.  We will be demonstrating how we can use droplet evaporation measurements simultaneous with growth rate measurements to check the chamber supersaturation in a future publication.
* * *
Now, the authors claim that "..., *typically no frost was observed on VSC walls for at least 6 hours*" (page 6 line 20).

In this case the question arises, how it was possible to conduct an experiment for 92 hours as described in the

section 3? Obviously, this would require multiple de-icing steps as described on page 6, lines 18-19, during which

the GC has to be disconnected from the VSC and reconnected to the second VSC with the VS set exactly to the

same temperature. Or was the GC just left connected to the VSCs resulting in no supersaturation, as implied by the

sentence "*During part A $S_a$ was not highly controlled but conditions were maintained such that −0.5% < $S_a$ <*

*0.5%.*"?

Our Reply:  For these long-period growth experiments frost would sometimes form on VSC walls but none was observed not on the GC walls (if frost were to appear on GC walls then experiments would be terminated and these walls cleaned before a new experiment could start).  CC2 is designed so that if one VSC does did start to grow frost then the sliding valve can be set in the opposite position to isolating this VSC from the GC.   With the valve in this new position the GC would be now experiencing vapor from a clean-walled 2nd VSC with its TEC set to the same VS temperature.  To remove frost off from a VSC wall we reduce the TEC temperature and pump out the chamber until the walls were clear of ice.  Now this newly clean-walled VSC is ready for use if frost were to form on the other VSC walls.
* * *
If the setup was build to study the ice crystal growth at atmospherically relevant conditions, it should be possible to set and maintain supersaturations up to 30%. Nothing in this manuscript tells me that this is feasible.

Our Reply: CC2 is designed to maintain much higher supersaturations than the experiments described here. At larger $S\_a$ we observe significantly larger growth rates. But this paper's focus is on the improved imaging capabilities and temperature stability of the instrument. The report of data sets made at different T conditions and at higher $S\_a$ will come later. The goal of the experiments reported here was not to explore the full range of T and S conditions but instead explore the differences in facet normal growth rates for crystals growing under the same conditions.
* * *
If calculation of the supersaturation based on the instrumental parameters is not possible, a calibration can be achieved by measuring diffusion growth or evaporation of a reference particle – droplet of a known solution. Apparently, the authors have done that: "...*we have in addition used droplet evaporation measurements done simultaneously with crystal growth measurements to give a direct, and independent, estimate of $S_a$*" (page 5 line 30). It's disappointing that this direct and independent estimate of the supersaturation is neither given nor discussed further. The only measurements of the crystal growth rate presented in the manuscript (Figure 6) are not compared with any other measurements or with theory; we learn that the growth rate can be fitted by a two-parameter power- law parameterization, but no further attempt of interpretation is given. Actually, even the fit parameters are not given or discussed, and the reader is informed "*A more detail discussion of these results is reported elsewhere*" leading to a reference (Swanson 2019a) that has a different title and dedicated to a different topic (I assume this is the reference to the paper in ACPD by the same authors. Actually, the reference to Swanson 2019b leads to nowhere). I have not been able to find any discussion of these results in the companion paper.

I am afraid I cannot recommend publishing the paper in its present form. It should be thoroughly revised aiming at providing a verifiable characterization of the apparatus under wide range of experimental conditions. If the supersaturation cannot be calculated from the instrumental parameters, it should be calibrated in a dedicated experiment with evaporation or condensation of inorganic solution droplet, and the results reported together with the theoretical model used for the simulations. I am not in a position to give advice on the chamber design, but perhaps it would be better to create well-defined wall boundary conditions in the VSC and CG chambers by covering walls with ice than relying on the absence of water adsorbed on the bare metal walls of the chamber.

1. Nelson, J. and B. Swanson (2019). "Air pockets and secondary habits in ice from lateral-type growth." Atmos. Chem. Phys. Discuss. **2019**: 1-51 doi: 10.5194/acp-2019-280.

Our Reply: As we mentioned above, we have rewritten this section to explain more clearly the limits to $S_a$ determination. For the Snomax crystal experiments $S_{vs}$ is now reported at the bottom of P. 9. To date there have been no experiments with a direct measurement of S near the crystal surface simultaneous with the growth rate measurements but we have made droplet evaporation measurements simultaneous with crystal growth measurements to obtain a direct measurement of $S_a$. But these measurements were not done during the Snomax crystal growth experiments as a more precise determination of $S_a$ is not required here. Crystals were grown simultaneously under the same thermodynamic conditions to observe differences in facet-normal growth rates under the same growth conditions. In future experiments when sequential measurements are to be compared then $S_a$ calibration will be an important part of our experimental procedure.